# Tracking changes in SARS-CoV-2 transmission with a novel outpatient sentinel surveillance system in Chicago, USA

Reese Richardson [1,2], Emile Jorgensen[2], Philip Arevalo[3], Tobias M. Holden[4], Katelyn M. Gostic[3], Massimo Pacilli[2], Isaac Ghinai[2], Shannon Lightner[5], Sarah Cobey [3] & Jaline Gerardin [4] ✉

Public health indicators typically used for COVID-19 surveillance can be biased or lag changing community transmission patterns. In this study, we investigate whether sentinel surveillance of recently symptomatic individuals receiving outpatient diagnostic testing for SARS-CoV-2 could accurately assess the instantaneous reproductive number $R(t)$ and provide early warning of changes in transmission. We use data from community-based diagnostic testing sites in the United States city of Chicago. Patients tested at community-based diagnostic testing sites between September 2020 and June 2021, and reporting symptom onset within four days preceding their test, formed the sentinel population. $R(t)$ calculated from sentinel cases agreed well with $R(t)$ from other indicators. Retrospectively, trends in sentinel cases did not precede trends in COVID-19 hospital admissions by any identifiable lead time. In deployment, sentinel surveillance held an operational recency advantage of nine days over hospital admissions. The promising performance of opportunistic sentinel surveillance suggests that deliberately designed outpatient sentinel surveillance would provide robust early warning of increasing transmission.

In the SARS-CoV-2 pandemic, the ability of public health agencies to monitor disease incidence and trends in transmission has formed a critical component of public health preparedness and response[1–4]. Worldwide, policymakers have implemented staged regional mitigation systems, where the progression of a region from one stage of mitigation policy to another is contingent upon certain indicators surpassing a given threshold or relative rate of growth[5–7]. The timeliness of mitigation measures is a decisive factor in their efficacy; delaying the implementation of mitigation measures can drastically increase prevalence, mortality, and the probability that healthcare systems are overwhelmed amidst a surge in transmission[8–12]. Thus, it is crucial that the indicators that inform these mitigation measures represent a timely and accurate measure of trends in infection prevalence.

Many common indicators of SARS-CoV-2 transmission are inherently biased or delayed. Incident cases, the fraction of diagnostic tests that return a positive result (test positivity rate, TPR), or any other metric based on diagnostic testing in the general population is subject to bias due to fluctuating access to, availability of, and demand for diagnostic testing. These factors vary across time, geography, age, and racial and ethnic groups, and the data needed to control for these biases is often unavailable[1,13–17]. The timeliness of data can also be hampered by long turn-around-times and delays in vendors' reporting of test results to health agencies[14].

Severe outcomes, such as COVID-19 hospital admissions, emergency department visits, and deaths, are more reliable indicators of community transmission[1]. However, hospital admission can lag infection by as much as two weeks, and deaths can further lag hospital

[1]Department of Chemical and Biological Engineering, Northwestern University, Evanston, IL, USA. [2]Chicago Department of Public Health, Chicago, IL, USA. [3]Department of Ecology and Evolution, University of Chicago, Chicago, IL, USA. [4]Department of Preventive Medicine and Institute for Global Health, Northwestern University, Chicago, IL, USA. [5]Illinois Department of Public Health, Springfield, IL, USA. ✉e-mail: jgerardin@northwestern.edu

**Table 1 | Delays associated with each indicator traditionally used for SARS-CoV-2 surveillance and the theoretical operational recency provided by outpatient sentinel surveillance**

| Indicator | Presentation date | Days from infection to presentation median (IQR) [source] | Days from presentation to report during study period | Days from infection to report (operational lag) |
|---|---|---|---|---|
| Cases | Date of specimen collection | 8 (4–14)[28,49] | ~2–3 | ~10–11 |
| Emergency department (ED) visits | Date of ED visit | 10 (7–14)[18,21] | ~1–2 | ~11–12 |
| Hospital admissions | Date of admission | 10 (7–14)[18,21] | ~5 | ~15 |
| Deaths | Date of death | 19 (13–27)[18,21,51] | ~1–30 | ~20–57 |
| Outpatient sentinel surveillance | Date of symptom onset | 5 (4–7)[21] | ~2 | ~7 |

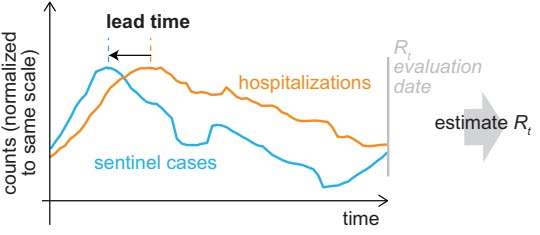
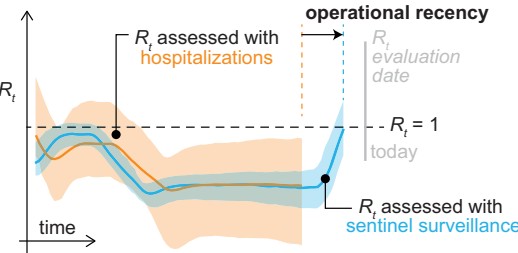

**Fig. 1 | Theoretical diagram of the instantaneous effective reproductive number $R(t)$ derived from hospitalizations (orange) and sentinel surveillance (blue).** Because symptom onset typically occurs sooner in the course of disease than a visit to the emergency department or hospitalization, sentinel cases can return more recent estimates of $R(t)$ than hospital-based indicators. With sufficient sample size, sentinel surveillance could also return more precise estimates of $R(t)$ (shaded regions show theoretical confidence intervals).

admission by another week (Table 1)[18–23]. Hospital- and death-based indicators are thus inherently limited in their ability to report very recent trends in transmission[24]. Furthermore, hospitalizations and deaths are mostly composed of older populations that are more likely to experience severe outcomes. As a result, these indicators provide more statistical power for discerning trends in older age groups than in younger age groups and could prove misleading if taken to represent the population at-large[1]. For instance, if a population-wide surge in disease burden is initially driven by an increase in transmission in younger age-groups, hospital-based indicators will not reflect this change until this increase in transmission propagates to the older age groups that drive admissions and ED visits. Additionally, if hospitalizations in older populations are greatly reduced by vaccination, signals of trends in community transmission derived from hospital admissions are further muddled.

Sentinel populations can be used to track changes in SARS-CoV-2 transmission in the general population and have previously been used or proposed for monitoring seasonal and pandemic influenza[25,26]. As long as testing criteria and sampling effort on the sentinel population are predefined and do not change with time, COVID-19 surveillance on the sentinel population should be less subject to selection bias than diagnostic testing in the population at-large. Although not ideal, even mobile or intermittently active sentinel surveillance sites may still be able to inform changes in transmission.

This study used data from patients tested at community-based testing programs operated by the Chicago Department of Public Health (CDPH) and the Illinois Department of Public Health (IDPH) to assess trends in SARS-CoV-2 transmission with minimal bias and lag in the United States (US) city of Chicago (Fig. 1). These community-based testing programs had been implemented to improve access to testing in underrepresented groups, and this study opportunistically reused the data to evaluate the potential utility of outpatient-based sentinel surveillance. Recently symptomatic individuals (onset within 4 days of test) were used as the sentinel population from which to estimate the instantaneous effective reproductive number $R(t)$, a measure of community transmission. In theory, this approach would provide operational recency over hospital-based indicators, since symptom onset occurs sooner after infection than hospitalization, allowing $R(t)$ to be estimated for more recent dates (Fig. 1, Table 1). Furthermore, sentinel surveillance would provide information on trends in younger populations than the hospitalized population, which, if changes in transmission occur first in younger populations, could result in trends in sentinel surveillance data leading trends in hospitalizations. The extent to which sentinel surveillance captured gold-standard hospital admission trends was evaluated and the lead time and operational recency of sentinel surveillance data over hospitalization data was assessed.

## Results

From September 2020 to June 2021, CDPH and IDPH operated a combined 10 static and 167 mobile community-based diagnostic testing sites in the city of Chicago (Fig. 2A). These testing sites targeted communities experiencing high COVID-19 incidence and demographic groups and geographic areas underrepresented in testing by other clinical providers[27]. Other than the Gately Park site, no static site was operational through the whole study period. Testing sites focused specifically on serving Hispanic/Latino residents because this population had the highest daily incidence of COVID-19 of any racial/ethnic group during the study period (Fig. 3). Diagnostic testing data from CDPH and IDPH community-based sites were re-analyzed in this study as outpatient sentinel surveillance. Of 324,872 total specimens collected during the study period, 21,406 were from Chicago residents with a valid recorded date of symptom onset, and 13,952 met the criteria to be sentinel samples (residing in Chicago with symptom onset date at most four days prior to specimen collection date, see Methods and Supplementary Fig. S1). Of the sentinel samples, 5401 were collected at CDPH-operated static sites (Fig. 2, sites a – h), 7,478 at IDPH-operated static sites (Fig. 2, sites i – j), and 1,073 at CDPH-operated mobile sites. The volume of sentinel samples collected each day fluctuated with the opening and closure of sites and decreased on weekends (Fig. 2B, C, Supplementary Fig. S2). Cumulatively across the

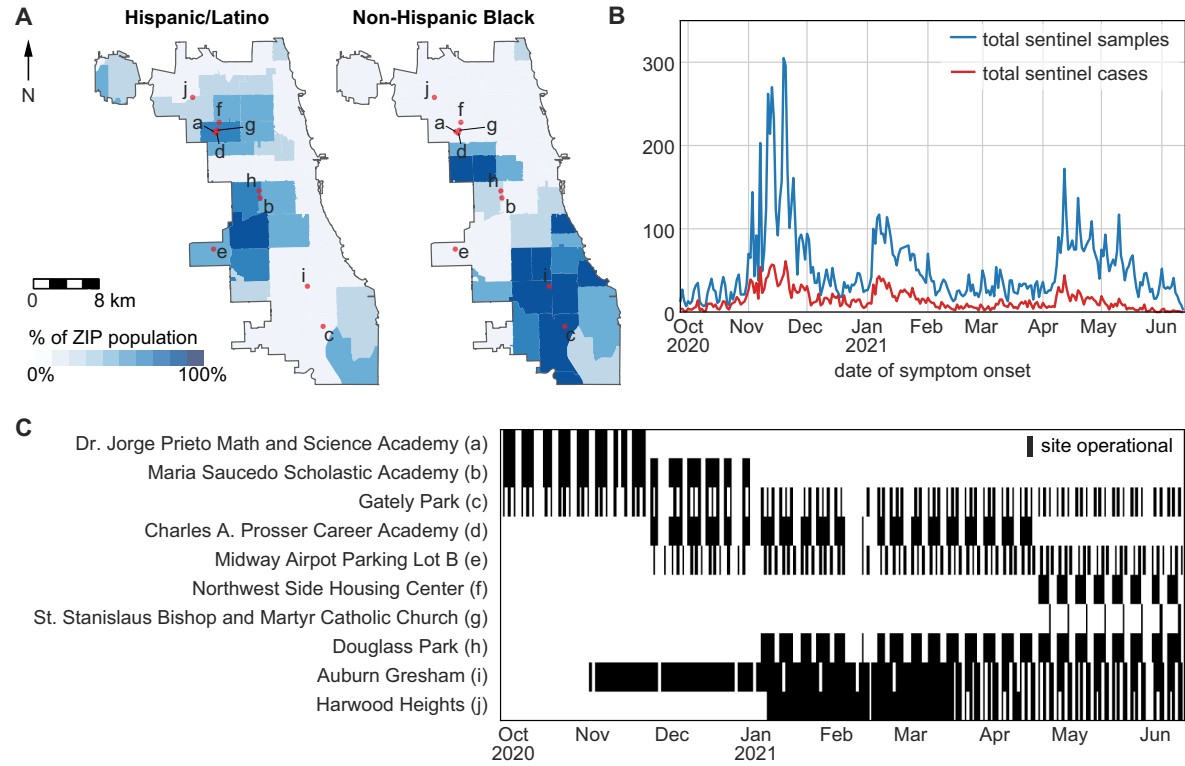

**Fig. 2 | Locations and volume of outpatient sentinel surveillance in Chicago from September 27, 2020, to June 13, 2021. A** Static sentinel testing site locations in Chicago. Letter labels correspond to site names in panel **C**. Colors indicate the percentage of each ZIP area's residents who are Hispanic/Latino (left map) or Non-Hispanic Black (right map). **B** Total sentinel samples (blue, *n* = 13,952) and sentinel cases (red, *n* = 3607) plotted by date of symptom onset. **C** Operating dates (black bars) of static sentinel testing sites.

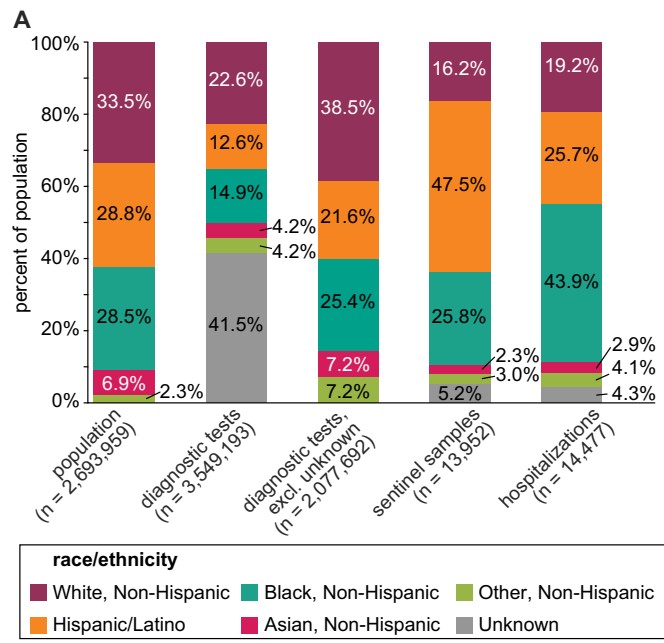

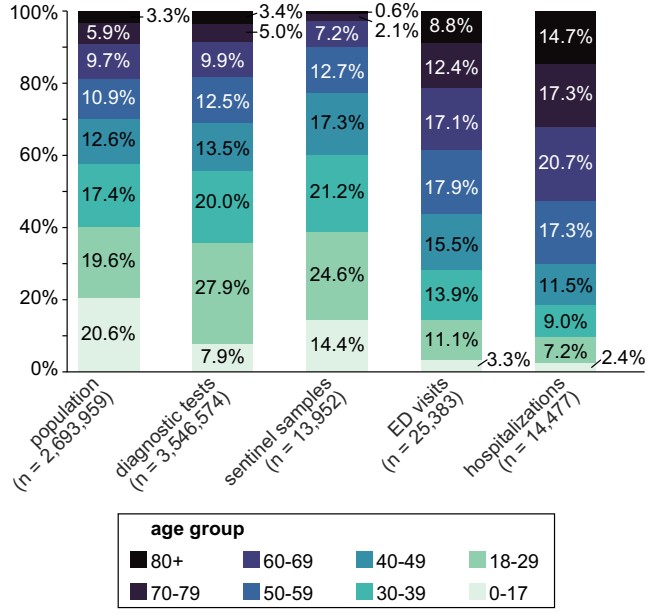

**Fig. 3 | Demographic characteristics of study population.** Broken down by **A** race/ethnicity and **B** age group for 2019 Chicago population estimates, all diagnostic tests, sentinel samples, COVID-19-confirmed emergency department (ED) visits, and COVID-19-confirmed hospital admissions during the study period. Race/ethnicity data was not available for ED visits. Three hospitalizations and 75 ED visits were of unknown age, and are excluded from **B**. Sentinel samples were a subset of all diagnostic tests (see Methods).

study period, 3607 sentinel samples returned a positive diagnosis (25.8%) and were considered sentinel cases (Fig. 2B, Supplementary Fig. S1). Across all 324,872 specimens collected at sentinel sites during the study period, 53,939 returned a positive diagnosis (Supplementary Fig. S3).

Compared with the general Chicago population and with all diagnostic tests performed during the study period, sentinel samples had a higher proportion of Hispanic/Latino residents (Fig. 3A). By proportion, the sentinel population was more Hispanic/Latino than COVID-19-confirmed hospitalizations, less non-Hispanic Black, and less non-Hispanic White. The age distribution of sentinel samples (9.9% greater than 60 years old) was younger than that of COVID-19-confirmed hospitalizations (52.7% greater than 60 years old, two-tail two-proportion $Z = -77.5$, $p < 0.001$) and COVID-19-confirmed emergency department (ED) visits (38.3% greater than 60 years old, $Z = -59.9$, $p < 0.001$), more closely resembling, but still younger than, the age distribution of the population at large (18.9% greater than 60 years old, $Z = -27.1$, $p < 0.001$) and the age distribution of all diagnostic tests (18.3% greater than 60 years old, $Z = -25.6$, $p < 0.001$) (Fig. 3B). Test positivity rates among sentinel samples was highest in ages 80+ and in Hispanic/Latino patients across the study period (Supplementary Fig. S4). Sentinel samples and all tests collected at sentinel sites were demographically similar, although a slightly higher proportion of sentinel samples were Non-Hispanic Black than among all tests collected at sentinel sites (Supplementary Fig. S5). Sentinel sites performed <10% of all diagnostic tests in Chicago during the study period and only 0.4% of all diagnostic tests in Chicago were also sentinel samples.

## R(t) Estimation

Trends in transmission were evaluated from time series derived from sentinel cases, sentinel test positivity rate (sentinel cases adjusted for testing volume, see Methods), general population cases, COVID-like Illness (CLI) emergency department visits (CLI ED), COVID-19-confirmed emergency department visits (COVID ED), CLI hospital admissions (CLI admits), and COVID-19-confirmed hospital admissions (COVID admits) by estimating the time-varying instantaneous reproductive number $R(t)$ from each data series (data series in Supplementary Fig. S6, $R(t)$ series in Fig. 4A). $R(t)$ was calculated with epyestim v0.1[28], a Python implementation of the method developed by Cori et al.[29]. $R(t) > 1$ indicates a growing epidemic and $R(t) < 1$ indicates a shrinking epidemic. The larger confidence interval for $R(t)$ estimates from sentinel cases toward the end of the study period reflects the decline in testing demand and lower number of sentinel cases collected in May-June 2021 (Fig. 2B). Assumed incubation periods and onset-to-presentation delays are detailed in Methods.

The agreement between $R(t)$ estimates derived from two data series was defined as the percentage of the study period when both median $R(t)$ estimates were ≥1.0 or both were <1.0. Agreement was highest between CLI ED, COVID ED, CLI admits, and COVID admits (Fig. 4B, Supplementary Fig. S7). $R(t)$ derived from sentinel cases agreed with $R(t)$ from COVID admissions on 84.7% of dates. Adjusting sentinel case counts by the volume of sentinel samples with the same day of symptom onset (sentinel TPR) did not improve correlation with other indicators, producing an $R(t)$ series with 68.2% agreement with COVID admits (Fig. 4B, Supplementary Fig. S7). $R(t)$ derived using sentinel samples as an indicator (analogous to CLI) produced slightly worse agreement with hospital-based indicators than $R(t)$ derived from sentinel cases. Three major inflection points occurred during the study period: the peak of the Fall 2020 wave (Nov 2020), the valley preceding the Spring 2021 wave (Feb 2021), and the peak of the Spring 2021 wave (Mar 2021). The dates of these inflection points in the sentinel cases $R(t)$ curve fall between 24 days behind to 11 days ahead of traditional indicators (Supplementary Table S1).

$R(t)$ derived from sentinel cases produced slightly better agreement with hospital-based indicators in the latter half of the study period (February–June 2021, Supplementary Fig. S8). Agreement between all $R(t)$ estimates worsened slightly when $R(t)$ was calculated with a seven-day smoothing window (Supplementary Fig. S9). When $R(t)$ was derived from indicators split into ages <60 and ≥60, agreement remained high for R(t) derived from sentinel cases age <60 and other $R(t)$ series derived from indicators reflecting ages <60. Agreement was lower between sentinel case $R(t)$ and $R(t)$ from other indicators reflecting ages ≥60 (Supplementary Fig. S10).

Of 21,046 specimens meeting all other criteria to be sentinel samples, 13,952 had a symptom onset date four or fewer days prior to their specimen collection date (65.2%) and 16,271 had a symptom onset date seven or fewer days prior to their specimen collection date (76.0%) (Supplementary Fig. S11). Varying the inclusion criteria for sentinel samples from symptom onset ≤3 days prior to specimen collection to symptom onset ≤7 days prior to specimen collection did not appreciably change retrospective agreement between $R(t)$ derived from sentinel cases and $R(t)$ derived from other indicators (Supplementary Fig. S12). During the first three months of deployment, variation in testing volume was accounted for by employing a subsampling technique wherein only sentinel cases from a random sample of 40 sentinel samples collected each day were considered. This technique did not improve retrospective agreement between $R(t)$ derived from sentinel cases and $R(t)$ derived from other indicators (Supplementary Fig. S13).

Evaluation of agreement between $R(t)$ series with a continuous metric (Spearman's $\rho$) closely qualitatively matched findings by our discrete agreement metric (Supplementary Figs. S7–S13).

## Lead time estimation

The lead time of sentinel cases over all cases, ED visits, and hospital admissions was evaluated by calculating cross-correlation functions between each case, visit, or admission timeseries in relation to the other timeseries. Changes in sentinel cases did not precede changes in any hospital-based indicators by any identifiable lead time (Fig. 4C). Changes in sentinel TPR returned positive lead times over cases and hospital-based indicators, albeit with low correlation and high uncertainty. Cases from the general population led COVID ED visits by about four days [lead time of 3 (−1, 8) days, peak $\rho = 0.932$]. CLI ED visits led CLI admits by about four days [lead time of 4 (0, 7) days, peak $\rho = 0.961$] and COVID ED visits led COVID admits by about three days [lead time of 3 (−1, 6) days, peak $\rho = 0.973$].

## Operational recency evaluation

To evaluate operational performance of sentinel surveillance with recently symptomatic patients, we first corrected for right-censoring of sentinel cases using epidemic nowcasting[30–35], drawing from empirical data collected during the study period to estimate the proportional completeness of recent data (Fig. 5A). We tested three models of proportional completeness, drawing from data from the last 30 days (past month retrospective), all previous dates in the study period (all-time retrospective), or all previous dates in the study period on the same day of the week as the date being nowcasted (day-of-week model). For each evaluation date in the study period, we applied each model of proportional completeness, then evaluated $R(t)$ (Fig. 5B). Nowcasting was not performed for hospital admissions due to inconsistent backfilling of hospitalization data across the study period. Where nowcasting can be applied to hospitalization data, counts are right-censored over a much larger window than with sentinel surveillance, engendering greater uncertainty in nowcasted estimates[30,31].

Operationally, complete estimates of $R(t)$ were available for a given date nine days earlier with sentinel surveillance data than with hospitalization data. With nowcasting, sentinel surveillance showed increases in $R(t)$ weeks before the same increase was registered by

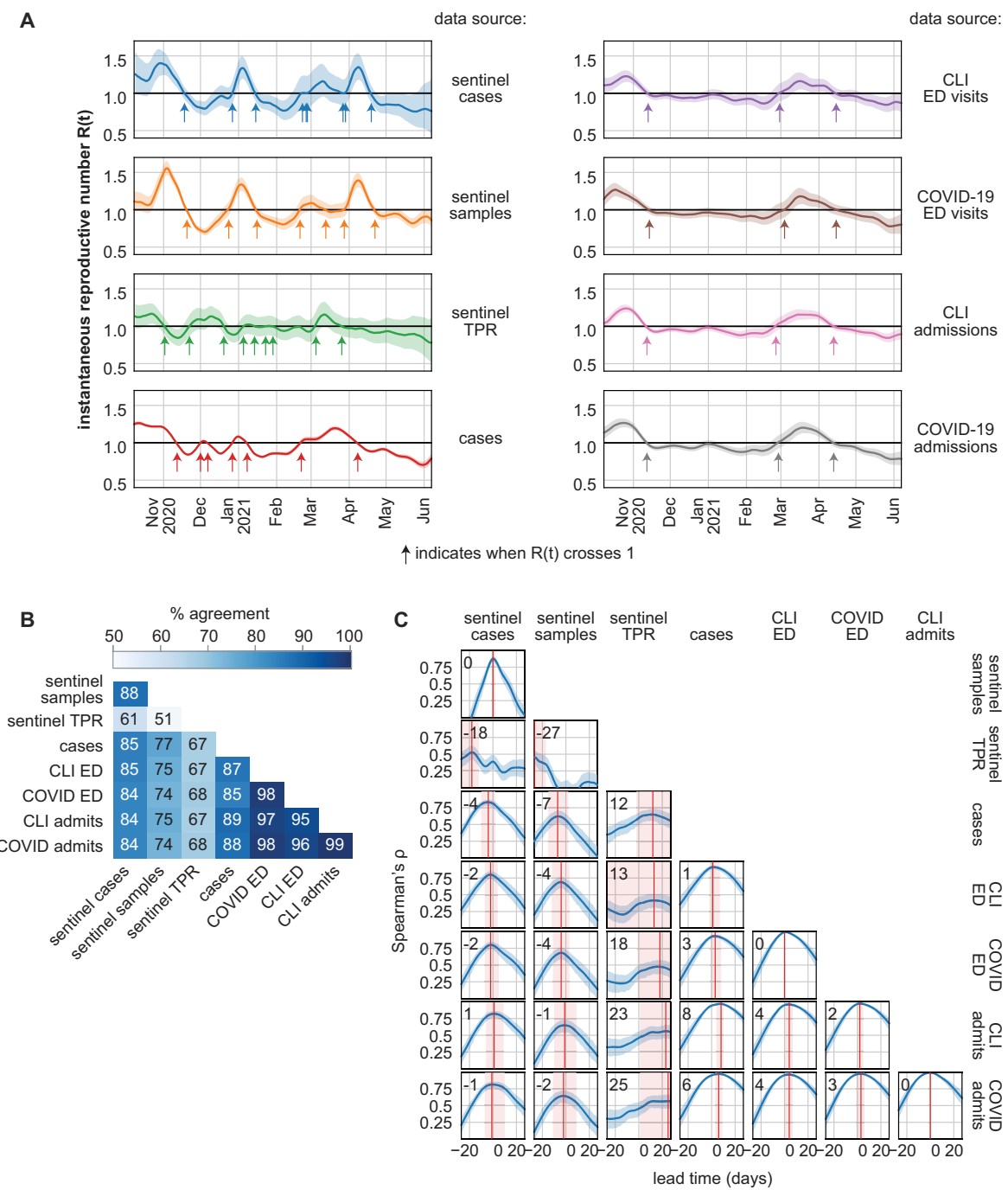

**Fig. 4 | Retrospective performance of sentinel cases at quantifying transmission and providing lead time compared with emergency department visits and hospital admissions. A** $R(t)$ calculated from eight types of surveillance data. Arrows indicate dates at which the median $R(t)$ estimate crosses 1.0. Solid lines indicate median estimate and shaded regions indicate the 95% confidence interval. **B** Similarity matrix of percent agreement between $R(t)$ series. Percent agreement is the percentage of dates when the median $R(t)$ estimates of two series are both ≥1.0 or both <1.0. Agreement was also assessed with Spearman's $\rho$ (see Supplementary Fig. S7). **C** Cross-correlation functions between eight types of surveillance data. Lead time indicates the number of days the series shown on the x-axis was displaced relative to the series on the y-axis. Positive lead time indicates that the x series leads the y series and negative lead time indicates that the x series lags the y series. Solid blue lines show nominal values of Spearman's $\rho$ and shaded regions indicate the 95% confidence interval about the nominal value from 1000 bootstrapped estimates. Red solid lines indicate the lead time at which maximum correlation is achieved; this lead time is noted in the upper left corner of each plot. Red shaded regions indicate an uncertainty bound for the lead time (see Methods). Seven-day smoothed time series are shown in Supplementary Fig. S6. TPR test positivity rate, CLI COVID-like illness, ED emergency department.

hospital data. For example, on an evaluation date of February 27, 2021, nowcasted sentinel case counts suggested that $R(t)$ had risen past 1.0 on February 20, 2021; for this particular increase in transmission, COVID-confirmed admissions only returned $R(t) > 1$ on evaluation date March 17, 2021, 18 days later. We calculated false positive and false negative rates of real-time $R(t)$ estimates by comparing against $R(t)$ values derived from uncensored sentinel case counts (Fig. 5C). Deriving $R(t)$ from censored counts frequently underestimated recent reproductive rates, with a false negative rate ($R(t)_{censored} < 1$ whereas $R(t)_{uncensored} \geq 1$) of 0.4 and a false positive rate ($R(t)_{censored} \geq 1$ whereas

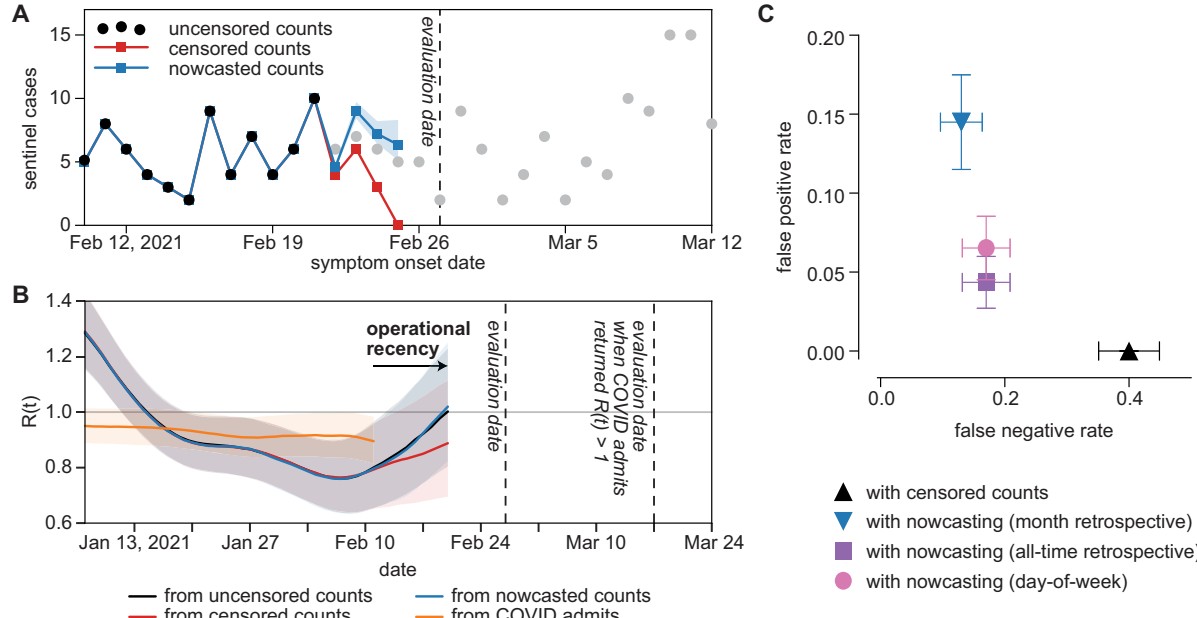

**Fig. 5 | Operational performance of sentinel surveillance. A** Sentinel case counts for a representative evaluation date of February 27, 2021. Black dots: sentinel case counts fully accessible on the evaluation date. Gray dots: sentinel case counts partially accessible or not yet accessible on the evaluation date. Red dots: right-censored sentinel case counts available on the evaluation date. Blue dots: median nowcasted sentinel case counts with a past-month retrospective model. Blue shaded region: 95% confidence interval for nowcasted counts. **B** $R(t)$ derived from uncensored sentinel cases, right-censored sentinel cases, nowcasted sentinel cases, and COVID-confirmed admissions for an evaluation date of February 27, 2021. Solid lines indicate median estimates of $R(t)$. Shaded regions represent 95% confidence intervals about the median. **C** False negative and false positive rates of $R(t)$ derived from right-censored and nowcasted sentinel cases when compared to $R(t)$ derived from uncensored case counts ($n = 238$ evaluation dates). Error bars represent the sample proportion ±1 standard deviation of the sample proportion.

$R(t)_{uncensored} < 1$) of 0.0. Nowcasting decreased the false negative rate at little expense to the false positive rate. Nowcasting with an all-time retrospective model returned a lower false positive rate than nowcasting with a past-month retrospective or day-of-week model.

## Discussion

This study used convenience data from community-based testing programs to opportunistically evaluate a potential outpatient sentinel surveillance system for COVID-19 based on the diagnostic testing results of recently symptomatic individuals. Although the daily volume of sentinel samples fluctuated substantially over the study period and the sentinel population was not demographically representative of Chicago's population at-large, $R(t)$ estimated from sentinel cases was in good agreement with $R(t)$ estimated from hospital data in the general population. Since the COVID-19 pandemic did not affect various racial and ethnic groups to the same extent in Illinois[17], and the sentinel population was more Hispanic/Latino than the general population, the small divergence between sentinel $R(t)$ and hospital-based $R(t)$ could indicate true differences in transmission dynamics between the sentinel and general populations. The larger proportion of Non-Hispanic Black Chicago residents in sentinel samples compared to all tests collected at sentinel sites (Supplementary Fig. S5) also reflects the potential for demographic bias during the selection of sentinel samples. However, the general agreement in sentinel $R(t)$ and hospital admissions $R(t)$ is impressive given that the sampling frame of the sentinel cases is more biased than hospital admissions (ignoring age bias) though less biased than most other types of surveillance.

Generally, adjusting sentinel cases for testing volume (sentinel TPR) did not improve agreement with other indicators. This could be reflective of broader biases in using TPR as a metric. For instance, in regular outpatient diagnostic testing, TPR can go up independent of incidence because of decreases in test availability or in test demand (e.g., during the snowstorms in Chicago in February 2021 when access

to testing was physically more challenging). This is because in periods where testing is more restricted, only those that are most likely to test positive (e.g., known prior contact or actively symptomatic) are likely to access a test. Conversely, TPR may also fall independently of incidence if a large volume of diagnostic tests is suddenly made available to a population with limited access to testing. It is possible that the lower agreement from TPR in the earlier part of the study period is due to more frequent changes in test site hours and locations (Fig. 1C, Supplementary Figs. S2 and S8).

Because symptoms are expected to develop an average of 5.5 days after infection and hospital admission occurs an average of 11.1 days after infection[18,21], the naïve expectation for retrospective lead time of trends in sentinel cases over trends in hospital admissions was 5−6 days. Retrospective cross-correlation analysis revealed no identifiable lead time between sentinel cases and hospital-based indicators. A combination of factors could contribute to this observation. First, the low number of daily counts obtained from sentinel surveillance adds significant noise, obscuring potential trends in lead time. Second, because delays between infection and symptom onset or between infection and hospitalization both follow skewed distributions (Supplementary Fig. S14), the median lead will be less than 5.5 days (i.e., less than the mean lead time). Third, although the use of smoothing windows was necessary in this analysis to remove day-of-week effects, smoothing blurs temporal changes in each indicator, which could complicate extracting trends through cross-correlation analysis. Finally, it is possible that the true distributions of days from infection to hospitalization and days from infection to symptom onset changed over the course of the study period with the arrival of different variants in Chicago. Estimates of lead time between an adjusted sentinel case time series (sentinel TPR) and hospital-based indicators, while positive, were highly uncertain. Sampling effort fluctuated substantially during the study period, and increasing the number of sentinel samples collected relative to the size of the general population would likely produce more precise estimates of lead time.

$R(t)$ estimates are sensitive to serial interval estimates. The serial interval distribution and delay distributions used were derived from empirical research conducted before the global emergence of the Delta variant (Pango lineage B.1.617.2), which is suspected to spread with a shorter serial interval than the strains in circulation at the time of these studies[36,37]. For the two-week period ending on June 5, 2021, the CDC estimated the national proportion among incident infections of all variants carrying a L452R spike protein substitution to be 15.3%, suggesting that the Delta variant's impact on the serial interval was likely minor across the study period[38]. The serial interval is also expected to change with implementation of non-pharmaceutical interventions (NPIs), which included back-and-forth impositions and relaxations of mask mandates and indoor dining restrictions during the study period[39]. However, the actual extent to which NPIs changed the behavior of Chicago residents is unknown, and it is unclear how to include changes in NPIs in estimates of the serial interval. $R(t)$ estimates are also sensitive to delay distribution estimates, and erroneous assumptions in the delay distribution of one indicator can harm the accuracy of pairwise comparisons of $R(t)$ and other indicators. For instance, if trends in sentinel cases and admissions were identical, but the assumed distribution from infection to symptom onset (the delay distribution used for sentinel cases) were too short, increases in $R(t)$ derived from sentinel cases would appear later in time than increases in $R(t)$ derived from admissions.

The timeliness and accuracy of recent admission and ED visit data is limited by the completeness and frequency with which individual hospitals and hospital systems report their data to public health agencies. Many COVID-confirmed hospitalizations are only reported several weeks after admission; for instance, three months after the conclusion of the study period, 6294 COVID-confirmed hospitalizations were newly recorded for dates during the study period. Sentinel surveillance with a trusted set of outpatient diagnostic testing vendors would circumvent this issue, ensuring that case counts are complete as soon as test results are returned and that the most recent estimates of epidemic growth are timely and accurate.

The operational recency advantage of sentinel surveillance was apparent during deployment in Chicago. Acute care hospitals in Illinois report emergency department and inpatient visit data to IDPH daily. Once per week, IDPH matches the hospital patients to the COVID cases recorded in the Illinois National Electronic Disease Surveillance System (I-NEDSS) and sends the results for Chicago residents to CDPH. Typically, CDPH is able to analyze the hospital data two days after the match is performed. As a result, hospital admissions are developed once per week and with about 5 days delay (e.g., a dataset made available on Friday would only include hospital admissions up to the preceding Sunday). On the other hand, complete data from sentinel testing usually became available after about two days (e.g., a dataset made available on Friday would include test results with specimen collection dates up to the preceding Wednesday). Thus, on a typical evaluation date, the most recent estimates of $R(t)$ from hospital admissions were for 16 days prior, whereas the most recent estimates of $R(t)$ from sentinel surveillance were for seven days prior. These reporting lags added to the intrinsic lag between symptom onset and hospitalization to create an aggregate advantage in operational recency of nine days for sentinel surveillance over hospital admissions.

IDPH submits de-identified information more frequently to the National Syndromic Surveillance Program (NSSP) which CDPH can access through the Electronic Surveillance System for the Early Notification of Community-Based Epidemics (ESSENCE). However, Chicago residents and non-Chicago residents cannot be disambiguated in this data, as it is de-identified and aggregated at the hospital level. If Chicago-resident-only data were available on a more frequent basis such that the reporting delay were only 1–2 days, as opposed to 5 days, outpatient sentinel surveillance would retain an operational recency advantage of about 5–6 days.

The operational recency of sentinel surveillance can be compromised by atypical operational delays that do not affect other indicators. During the study period, although typical wait times for complete sentinel testing data were around two days, delays in test turnaround time occasionally further extended this wait. In the most extreme instance, in February 2021, inclement weather closed several sentinel testing sites and prolonged the delivery of specimens to vendor laboratories, causing wait times for sentinel data in excess of seven days.

Deployment of outpatient symptomatic sentinel surveillance relies on robust and consistent collection of symptom data (including date of symptom onset) across time and across sites. Despite federal guidance[40], such symptom data has been very poorly collected at US outpatient diagnostic testing sites. In this study, collection of symptom data was not consistent between community-based testing sites and thus symptomatic individuals could only be identified by their own reporting of a symptom onset date rather than by meeting a consistent definition of "symptomatic". Stringent standards for collection of symptom data should be established prior to and enforced during deployment of this method of sentinel surveillance. In settings where symptom status is well-collected but symptom onset date collection is relatively incomplete, date of symptom onset could feasibly be imputed for missing values[41]. Even with these limitations related to the sentinel population chosen for this study, estimates of community transmission derived from sentinel cases approximated those of established, hospitalization-based indicators – with a population based sample and standardized collection of symptom information, such as the UK's Office for National Statistics Infection Survey[42], the performance and value of this sentinel surveillance model may be enhanced. This approach could be aided by voluntary home-based, app-enabled symptom data collection, such as with the UK's ZOE Health Study[43], or an app that complements at-home antigen-based testing and inquires about symptom onset. Integrating sequencing and virus subtyping into a sentinel surveillance with consistent symptom data collection would also rapidly provide information on symptom presentation of emerging variants. Such a sentinel surveillance system would be continuously useful for its advantage in operational recency but would be especially useful in scenarios of low outpatient testing availability (as symptomatic individuals would be far more likely to seek a test) and in scenarios where mass vaccination lowers rates of severe outcomes and limits the statistical power of hospital-based indicators.

In Chicago, the low volume of sentinel samples ultimately limited the precision of trends estimated from sentinel surveillance. However, that even a low-volume, unrepresentative, and opportunistic outpatient sentinel surveillance performed well strongly suggests that a deliberate sentinel surveillance system, with high testing volume, routine reporting of date of symptom onset, and representative sampling of outpatient providers, would provide robust early warning. With sufficient sentinel sampling volume and consistency of site availability to residents over time, neighborhood-level $R(t)$ estimations should be possible. Under conditions of exponential growth, even 1–2 weeks' early warning could save lives.

## Methods

### Data collection for sentinel surveillance

Chicago is an urban area of 2.7 million people located in the central US state of Illinois[44]. From May 13, 2020, CDPH and IDPH operated community-based SARS-CoV-2 diagnostic testing sites throughout Chicago. These sites were primarily intended to increase access to diagnostic testing among communities disproportionately affected by COVID-19 and those with the least access to diagnostic testing through other providers[27]. This study focuses on the period from September 27, 2020, to June 13, 2021, when these sites held consistent hours and reliably collected information on symptoms. In this period, CDPH-

operated static sites at eight locations (Fig. 2, sites a–h) and IDPH operated sites at two locations (Fig. 2, sites i–j). CDPH also held 167 single-day mobile testing site events during this period, during which testing vans were positioned outside a community venue for 4–6 h. Hours of operation varied by test site and day of week. In March 2021, IDPH sites moved from offering testing all seven days a week to just three days a week. All sites, mobile and static, solely offered anterior nares molecular (PCR) diagnostic tests. All individuals receiving a test were asked to report recent symptoms and provide the date of symptom onset. Those testing at IDPH sites were asked to report the presence or absence of symptoms from COVID-19 symptom list from the Centers for Disease Control and Prevention (CDC)[45]. Those testing at CDPH sites were asked only to report the presence or absence of any symptoms without reference to any list of expected symptoms of COVID-19. Due to this discrepancy in the collection of symptom data, symptom status for sentinel surveillance was determined by the presence of absence of a symptom onset date. All Chicago residents reporting symptom onset within four or fewer days of their specimen collection date were included in the sentinel samples. Sentinel cases were defined as sentinel samples with a positive test result (Supplementary Fig. S1). Specimens were collected at testing sites and transported to an off-site laboratory for processing via PCR. Testing vendors notified patients of results electronically as soon as results were available. During the study period, the median turn-around time from specimen collection to result notification was 2 days, with 95% of tests turned around between 1 and 4 days. Data was pulled on July 6, 2021.

### Other data sources

COVID−19-confirmed hospital admissions among Chicago residents, COVID-like illness (CLI) emergency department visits among Chicago residents, COVID−19-confirmed emergency department visits among Chicago residents, and all cases among Chicago residents were obtained from the City of Chicago Public Data Portal[46,47]. CLI admissions among Chicago residents were obtained from IDPH on August 25, 2021. Demographic data by ZIP code and citywide were obtained from the 2019 U.S. Census Bureau American Community Survey through the City of Chicago Public Data Portal[44].

Data on diagnostic tests in the general population were obtained from the Illinois National Electronic Disease Surveillance System (I-NEDSS) database and included PCR and antigen tests, but not serological tests, performed in Illinois on Chicago residents between September 27, 2020, and June 13, 2021. Data were reported as daily total tests by single year of age and race/ethnicity. Ages ranging from 0 to 116 years were considered valid, and others were reassigned null values. Racial/ethnic values included Hispanic/Latino, White non-Hispanic, Black or African American non-Hispanic, Asian non-Hispanic, Native Hawaiian or Other Pacific Islander non-Hispanic, American Indian or Alaskan Native non-Hispanic, Other non-Hispanic, and Unknown. Due to small sample sizes, data rows indicating Native Hawaiian or Other Pacific Islander, American Indian or Alaskan Native, or multiple racial/ethnic categories were reassigned as Other non-Hispanic.

### R(t) estimation

$R(t)$ was estimated from case time series with epyestim v0.1[28], a Python implementation of the method developed by Cori et al.[29]. All $R(t)$ estimates used epyestim's default SARS-CoV-2 serial interval distribution, derived from Flaxman et al.[48], and a final rolling average window of 14 days (r_window_size = 14). Without knowledge of the date of a case's actual date of infection, this $R(t)$ estimation method attempts to infer the date of infection for each case based on the case's date of presentation (e.g., date of symptom onset, date of specimen collection, date of hospital admission, etc). To estimate the date of infection, epyestim uses a "reporting delay distribution", which represents an estimated distribution of days between infection and this presentation date (defined in Table 1 for each data type). For sentinel data, the reporting delay distribution is the time from infection to symptom onset, which was approximated with a gamma distribution with shape factor 5.807 and scale factor 0.948 (mean 5.51 days)[21]. The time from symptom onset to hospitalization or emergency department visit was approximated with a gamma distribution with shape factor 1.104 and scale factor 5.074 (mean 5.60 days) (Supplementary Fig. S14)[18]. The reporting delay distribution for hospital admissions and emergency department visits represents the total time from infection to hospitalization and was thus modeled as the sum of the infection-to-onset and onset-to-hospitalization distributions using a gamma distribution with shape factor 3.667 and scale factor 3.029 (mean 11.11 days) (Supplementary Fig. S14). For cases in the general population, the reporting delay distribution represents the time from infection to test (date of specimen collection). This was modeled with epyestim's default reporting delay distribution, derived from a convolution of the incubation time distribution and the onset to test distribution derived from Brauner et al. (mean 10.33 days) (Supplementary Fig. S14)[49].

To adjust for changing sentinel testing volume over the study window, an additional sentinel time series, sentinel test positivity rate (sentinel TPR), was calculated for each date of symptom onset by dividing the total number of sentinel cases by the total number of sentinel samples, multiplied by the average number of daily sentinel samples across the study window (53.9 samples/day).

### Lead time estimation

To estimate the lead time of one metric over another, pairwise cross-correlation functions were made between counts timeseries of sentinel cases, sentinel TPR, cases in the general population, CLI emergency department visits, COVID-19-confirmed emergency department visits, CLI hospital admissions and COVID-19-confirmed hospital admissions. A seven-day centered moving average was applied to each raw time-series to eliminate day-of-week effects. For each cross-correlation function, one timeseries was iteratively displaced by −25 to 25 days and Spearman's $\rho$ was calculated between the displaced and non-displaced timeseries. All calculations of Spearman's $\rho$ were supplemented with 1000 bootstrapped estimates to produce 95% confidence intervals of $\rho$ at each lead time. The lead time at which Spearman's $\rho$ achieved its maximum was considered the point estimate of lead time for that comparison. The uncertainty in the lead time was estimated by taking the minimum and maximum lead times at which the 97.5th percentile of bootstrapped estimates of Spearman's $\rho$ were greater than the maximum nominal correlation.

### Operational recency evaluation and nowcasting

Epidemic nowcasting[30–35] was used to correct for recent under-reporting in sentinel case counts. For each date $t$, we estimated the proportional completeness of counts on dates $t − 5$, $t − 4$, $t − 3$, and $t − 2$ at the time of evaluation to inflate the sentinel case counts on those dates prior to calculating $R(t)$. Proportional completeness was estimated from a retrospective window of one month, all time, or by day-of-week. For the past-month retrospective model, the retrospective window used was the 30 dates of symptom onset immediately preceding $t − 5$. For the all-time retrospective model, the retrospective window used was all dates in the study window preceding $t − 5$. For the day-of-week retrospective model, the retrospective window used was all dates in the study window preceding $t − 5$ that shared the same day of the week as the date being nowcasted. If the censored count on any date to be nowcasted was zero, a pseudo-count of 1 was used. Each proportional completeness model was evaluated for every potential evaluation date in the study period. On each date $t$, right-censored counts and all three sets of nowcasted counts were used to estimate $R(t)$. These estimates of $R(t)$ were compared to estimates of $R(t)$ derived from uncensored

counts. Instances where the most recent $R(t)_{nowcast} < 1$ but the most $R(t)_{uncensored} \geq 1$ were counted as false negatives. Instances where the most recent $R(t)_{nowcast} \geq 1$ but the most recent $R(t)_{uncensored} < 1$ were counted as false positives.

## Ethical review

Research methods were performed in accordance with relevant guidelines and regulations. The Northwestern University Institutional Review Board has ruled that this study does not constitute human subjects research.

## Reporting summary

Further information on research design is available in the Nature Research Reporting Summary linked to this article.

## Data availability

Restrictions apply to the availability of sentinel surveillance data and individual-level diagnostic tests from I-NEDSS, which contain identifiable private health information. Interested parties should complete CDPH (https://www.chicago.gov/city/en/depts/cdph/provdrs/health_data_and_reports/svcs/data-request-form.html) or IDPH (https://dph.illinois.gov/content/dam/soi/en/web/idph/files/forms/formsoppsdischarge-data-request-form.pdf) data request forms to inquire about access to the I-NEDSS database and data use agreements; IDPH and CDPH will determine access on a case-by-case basis. Public data on cases, testing, ED visits, and hospital admissions are available from CDPH's Public Data Portal (tests: https://data.cityofchicago.org/Health-Human-Services/COVID-19-Cases-Tests-and-Deaths-by-ZIP-Code/yhhz-zm2v, cases and hospitalizations: https://data.cityofchicago.org/Health-Human-Services/COVID-19-Daily-Cases-Deaths-and-Hospitalizations/naz8-j4nc, ED visits: https://data.cityofchicago.org/Health-Human-Services/COVID-Like-Illness-CLI-and-COVID-19-Diagnosis-Emer/qwib-edaw).

## Code availability

All code used for data analysis is available at https://github.com/numalariamodeling/chicago_sentinel_surveillance (https://doi.org/10.5281/zenodo.7041699)[50].

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

## Acknowledgements
RR was supported by a grant from NIGMS (T32 GM008449). T.H. and J.G. were supported by the Peterson Foundation Pandemic Response Policy Research Fund. JG was supported by the NIGMS MIDAS COVID-19 Urgent Grant Program (MIDASNI2020-4).

## Author contributions
Conceived the study: R.R., E.J., I.G., S.L., S.C., J.G. Collected data: E.J., M.P., I.G., S.L. Analyzed data: R.R., E.J., P.A., T.H., K.G., S.C., J.G. Wrote the manuscript: RR. All authors have read and reviewed the final manuscript.

## Competing interests
The authors declare no competing interests.
