## [Peer Review File · Nature Communications]

Tracking changes in SARS-CoV-2 transmission with a novel outpatient sentinel surveillance system in Chicago, USAREVIEWER COMMENTS

Reviewer #1 (Remarks to the Author):

Summary

In the paper: "Tracking changes in SARS-CoV-2 transmission with a novel outpatient sentinel surveillance system in Chicago, USA," the authors develop a sentinel surveillance system for COVID-19 for real time tracking of the pandemic dynamics and analyze results concerning the utility of the surveillance program against more traditional epidemiological data streams. The paper is extremely well-written, the analyses are reasonable and robust, and I believe their conclusions are well supported by the data. The major conclusions appear to be that the biases in the sentinel surveillance makeup don't seem to bias the epidemiological estimates, and that operational delays make it so their surveillance program has a 9 day lead time to other epidemiological data sources. However, the paper left me wanting a bit more as it feels like the authors could squeeze a bit more out of the incredible dataset they have compiled. My main comments involve a few suggested avenues to explore the data that could provide extra support for the novelty and utility of their implemented surveillance system (maybe the authors did explore these areas to no avail already, but I think including the negative results would be useful). I think these analyses could dramatically increase the impact of the article, though the article, in presenting the novel surveillance system and the current analyses, is likely publishable as is (pending addressing at least my major comment).

Major comments

The sentinel surveillance program operated in Chicago is clearly an important and useful contribution to tracking the state of the pandemic. However, one of the key benefits noted by the authors is that it provides an operational lead time of about 9 days compared to the more traditional epidemiological features. It strikes me that this operational delay isn't necessary in those other epidemiological metrics. For example, couldn't an alternative to setting up a sentinel surveillance system be to produce the hospital admissions estimates in a more timely manner? HHS is currently providing hospital admissions data, albeit on a different geographic resolution than the current study, with only a day or two delay. I think it would be worthwhile for the authors to discuss the possibility of shortening the delays in hospital admissions data and how the sentinel surveillance program would continue to be useful if that occurred (e.g. if new results are discovered in response to my subsequent comments about lead time and geographic resolution).

Analysis suggestions

I agree with the authors that low sample size and noise in the sentinel case samples may be obscuring the lead time they might provide against hospital admission, though (<https://www.pnas.org/doi/10.1073/pnas.2111870119>) suggests there may be little lead time between case counts and hospital admissions. One thought in looking at the data is that the sentinel sample counts may be a good indicator in themselves (not just the positives). Have the authors explored that as a potential predictor? It seems similar to CLI, in that it captures a metric of symptomatic infections in the region, even if individuals don't ultimately test positive, and could provide a useful indicator to compare in Figure 4 and the supplemental analyses.

It is interesting that the case counts provide little lead time for the hospital admissions, but I believe the authors could do a bit more to more fully explore the data. For example, the critical time periods for public health decision making generally fall to detecting a pandemic surge and detecting the peak of a pandemic. Do the sentinel samples/cases provide lead time to these specific time periods over the course of the pandemic? One way to investigate this would be to identify the timing of the inflection points in the time-series (start and end of detected exponential growth, derivative changes, etc.), and see how the timing compares across the time series for these specific events rather than comparing the cross correlations across the full time-series.

Other studies have found strong geographic determinants for risk during the pandemic, and it appears that the authors have geographically stratified data. Are there any interesting features

looking at the correlations across testing locations or comparing the sentinel geographic trends with the geographic trends in hospital admissions/other metrics?

The authors mention and show the demographic data in Figure 3, but it appears that these data aren't used during the analyses. I think it would be useful to provide some results along these axes. For example, is test positivity highest in one race/ethnicity or age group compared to the others? The authors mention other studies that identified those trends, but it would be useful to see if similar results are obtained from the sentinel surveillance program.

I think it would be useful to see all of the time-series of the indicators aligned with one another in some plot (maybe a supplemental figure). That would allow readers to get a better handle of the raw data underlying the R_t and correlation results.

Reviewer #2 (Remarks to the Author):

This is a very well presented and thoughtful study detailing the advantages of using sentinel surveillance of outpatients for epidemic inference, which (if connected correctly with fast action policy making) could have given policy makers vital extra days to evaluate epidemic trajectory and assess current intervention suitability. I have a couple of what I'm calling major comments, but this is mainly driven by interest and I think would add to the paper overall but are not flaws of the current methods. These relate to adding a continuous assessment of sentinel data rather than discrete ($R_t > 1$ or < 1), which is also very useful for policy decisions from experience, and also exploring potential drivers of inaccurate R_t estimates.

Comment 1: In the operational recency evaluation (by the way this is really lovely phrasing for this - really clear) and in the similarity matrices used in Figure 4B and in the SI, you currently evaluate each data type by comparing if it correctly infers $R_t > 1$ or $R_t < 1$ in agreement with another data source. While it is important to know directionality of the epidemic, having a good estimate of continuous R_t is also important, as it also directly helps predict for example how many days until hospital beds become limited, which is helpful for planning and resource management. As a result adding a continuous similarity matrix (mean difference between median R_t estimates across all dates for example) would be helpful to know how accurate the sentinel data source is.

Comment 2: Over half of the sentinel samples are from static sites, however, the contribution of each of the static sites to total sentinel samples changes over time, e.g. Harwood Heights contributes only in 2021. Is this because the static sites only became active at certain points in time? Or because of where the epidemic was spreading? Please add this comment as well as perhaps plotting in the SI the incidence of cases in each of the static sites over time in comparison to the Chicagoland incidence series as a whole.

Minor Comments:

Line 48-55: This is an important point that hospitalisations and deaths are more likely to be older individuals. I am not sure I agree with the phrase "better represent" here though and how it connects to being uninformative of transmission patterns in younger ages. The issue is that because severity increases with age, you have more statistical power to discern transmission in older ages than you would in younger ages just due to sample size. Practically speaking, you will not be able to detect changes in younger populations, but likely because there are not enough admissions to be statistically confident in a change rather than it being uninformative. Perhaps change to express that if transmission trends are different in younger populations that it will be harder to detect from hospitalisations and deaths due to the age gradient in severe outcomes.

Figure 1: Really clear graphic. Just one comment - Perhaps add to legend that today is the R_t evaluation date (or change in figure) to get it to align with Figure 5.

Table 1: Could you add "Deaths" as an indicator and associated delays. You talk about deaths in your introduction and reference this table so I was surprised to see deaths not in here.

Line 88: After reading the introduction I was surprised to read that the majority of the testing sites were mobile, because after reading the introduction this surprised me as in lines 56-50 it says "testing criteria and sampling effort on the sentinel population are predefined and do not change with time". Perhaps just worth mentioning in the introduction that sentinel sites despite being mobile may still be informative of transmission.

Figure 2A: It is quite hard to work out percentages from this plot when you have both Hispanic/Latino and Non-Hispanic Black populations, i.e. where there is a mix of the blue and red (e.g. Auburn Gresham i) in 2A). If the plot is trying to just convey that there are regions with high % Hispanic/Latino, high % Non-Hispanic Black, low % either and mixed regions, maybe a 4x4 bivariate color scale may be clearer and you can then plot the full 4 by 4 color scale.

Line 103: Can also add reference to Figure 2B here which shows the relationship between sentinel cases and sentinel samples.

Line 112-119: Were any of these statistically different to the population as a whole? Given the denominators it seems likely that they are, but it would be good to add a statistical test here to convey that the sentinel samples are testing a significantly different proportion of the population.

Line 130: Please change CLI to COVID-like Illness (CLI) as this is the first time it has been used.

Figure 4: Perhaps it would be helpful to show as another column here or linked to another SI figure, the count data underpinning the Rt in Figure 4A. This might help guide the reader a little bit more by the time they get to Figure 4C in terms of understanding the lead times etc. (It took me quite a few times to read through this section to be able to understand all the information that is contained in Figure C, which I think would have been much more obvious with the count data also shown.)

Figure 5: Could you add that shaded regions in B are 95% CI (presumably). Similarly what are the error bars in Figure 5C.

Lines 192-197: The text here is very informative for explaining the advantages and makes the message very clear. It is hard though to directly relate this to Figure 5. Perhaps it would be helpful to have COVID admits shown here.

Discussion: Really well written and good paragraph discussing the lack of lead time between sentinel cases and hospital-based indicators.

Line 300: UK Zoe app also a good reference here showing that actually, UK population provided symptom data has been very informative of trends and has aligned well with more formal structures such as the UK ONS.

Reviewer #3 (Remarks to the Author):

The authors of the manuscript "Tracking changes in SARS-CoV-2 transmission with a novel outpatient sentinel surveillance system in Chicago, USA" describe the setup of a primary care outpatient targeted monitoring system to specific areas and compare the usefulness and added value with comprehensive systems as well as severe disease monitoring systems.

The study is well conducted and provides good insight as well as proposes strategies to focussed sentinel monitoring away from comprehensive testing and monitoring strategies.

A few comments to clarify some points are added below:

In the study it is assumed that the risk of hospitalisation is equally distributed across all age groups to be able to compare between primary and secondary care. However, the risk to be hospitalised increases significantly with increasing age and with the higher proportion of younger age groups in the sentinel the risk to be hospitalised might not be comparable to data from hospital settings. This could add bias to the results when comparing different age groups or risk groups across the different data sets. Have you tried to just use the 60+ age group for the comparison? Or use an age stratification? This might help to account for this risk difference in the

sampled populations.

It is unclear if there might be a bias by the inclusion of those with known symptom onset in the sentinel group. Have you tried a sensitivity analysis that compared the included and excluded groups in terms of demographics but also on test performance/positivity including over time? Are data from people not included into sentinel due to lack of symptom information included in the overall dataset? Are the sentinel data included in the total case data? Please add for clarity.

It is shown that sentinel test positivity performed worse than just pure case count, which is quite surprising. While simple case count is dependent from number of tested people (and proportion testing positive), this might be influenced by different testing strategies over time. Could you give some more details about this (e.g., in the supplement). This has not been discussed yet and seems remarkable as TPR is a commonly used indicator (for week positivity in flu surveillance systems). Have you tried using weekly data with positivity? Why do you think TPR does not describe the epidemiological situation well although having denominator data available? Could you reflect a bit more on this please? Fig S3 shows a much better performance of TPR in the later period than in the earlier. This is an interesting finding and could be addressed in the discussion.

It has not been described if this sentinel system would become permanent or just served temporarily. No outlook has been added in terms of the usefulness of (temporary) sentinel systems to monitor the situation of SARS-CoV-2 and also include other respiratory viruses. Could you please elaborate a bit more on this how such a system could be used and embedded to complement other systems and when it would be most beneficial.

I'm no expert in nowcasting and cannot assess the suitability of this approach although the results are promising. How well and timely could such nowcasting be implemented in ad hoc systems or newly established sentinel system and how much has nowcasting been used for public health decision making during the time of the sentinel system implementation in Chicago.

It's said that a PCR was performed for sentinel sites, but it is unclear if the tests have been at sentinel sites or had to be transported to lab facilities and how the reporting to patients and health authorities had been conducted. Could you please add some more details?

Sentinel systems for influenza are considered the gold standard as they are set up to be representative of the population to monitor the activity, spread and the circulating viruses in a timely manner. Sentinel flu viruses are sent to National Influenza Centres where viruses are subtyped, genetically and antigenically characterisation. This provides data for vaccine composition and delivers subtype information that is not available for most other specimens tested in non-sentinel diagnostic laboratories (at least in Europe). Sending specimens to central reference laboratories delays results which might not be relevant for influenza, but for SARS-CoV-2 particularly when a new variant emerges that needs to be closely monitored. The study does not mention any specific analysis of SARS-CoV-2 variant viruses among the positive tested specimens. It should be discussed how to integrate virus characterisation in such a sentinel system e.g., per specific drop-out PCR, sequencing, or more sophisticated analyses e.g., requiring virus culture.

Many thanks,
Cornelia Adlhoch

Notes to all reviewers:

Some agreement rates in pairwise comparisons of $R(t)$ traces (e.g. in Figure 4B) will differ slightly in this version compared to our original submission. The Cori method for $R(t)$ estimation relies on random sampling of input distributions to account for uncertainty in serial interval estimates. Because each iteration of the $R(t)$ estimation algorithm uses independent instances of bootstrap aggregation (bagging), each iteration will yield slightly different results. To aide reproducibility and consistency, we have now reset the seed of the algorithm's pseudorandom number generator before every instance of $R(t)$ estimation to ensure that identical inputs will always return identical results.

Additionally, we discovered that due to mislabeling of mobile sites, six sentinel samples were incorrectly labelled in-text as having been collected at static sites. In total, 5,401 sentinel samples were collected at CDPH-operated static sites (previously written as 5,407), 7,478 at IDPH-operated static sites, and 1,073 at CDPH-operated mobile sites (previously written as 1,067). The Results section now reflects the correct totals by site type.

REVIEWER COMMENTS

Reviewer #1 (Remarks to the Author):

Summary

In the paper: "Tracking changes in SARS-CoV-2 transmission with a novel outpatient sentinel surveillance system in Chicago, USA," the authors develop a sentinel surveillance system for COVID-19 for real time tracking of the pandemic dynamics and analyze results concerning the utility of the surveillance program against more traditional epidemiological data streams. The paper is extremely well-written, the analyses are reasonable and robust, and I believe their conclusions are well supported by the data. The major conclusions appear to be that the biases in the sentinel surveillance makeup don't seem to bias the epidemiological estimates, and that operational delays make it so their surveillance program has a 9 day lead time to other epidemiological data sources. However, the paper left me wanting a bit more as it feels like the authors could squeeze a bit more out of the incredible dataset they have compiled. My main comments involve a few suggested avenues to explore the data that could provide extra support for the novelty and utility of their implemented surveillance system (maybe the authors did explore these areas to no avail already, but I think including the negative results would be useful). I think these analyses could dramatically increase the impact of the article, though the article, in presenting the novel surveillance system and the current analyses, is likely publishable as is (pending addressing at least my major comment).

We thank the reviewer for their positive comments and have incorporated many of their analysis suggestions into the revised manuscript.

Major comments

The sentinel surveillance program operated in Chicago is clearly an important and useful contribution to tracking the state of the pandemic. However, one of the key benefits noted by the authors is that it provides an operational lead time of about 9 days compared to the more traditional epidemiological

features. It strikes me that this operational delay isn't necessary in those other epidemiological metrics. For example, couldn't an alternative to setting up a sentinel surveillance system be to produce the hospital admissions estimates in a more timely manner? HHS is currently providing hospital admissions data, albeit on a different geographic resolution than the current study, with only a day or two delay. I think it would be worthwhile for the authors to discuss the possibility of shortening the delays in hospital admissions data and how the sentinel surveillance program would continue to be useful if that occurred (e.g. if new results are discovered in response to my subsequent comments about lead time and geographic resolution).

Acute care hospitals in Illinois report emergency department and inpatient visit data to IDPH more or less daily. Once per week, IDPH matches the hospital patients to the COVID cases recorded in the Illinois National Electronic Disease Surveillance System (I-NEDSS) and sends the results for Chicago residents to CDPH. Typically, CDPH is able to analyze the hospital data two days after the match is performed. As a result, hospital admissions are developed once per week and with about 5 days delay. While the IDPH match takes advantage of access to personally identifiable information, IDPH does submit de-identified information more frequently to the National Syndromic Surveillance Program (NSSP) which CDPH can access through the Electronic Surveillance System for the Early Notification of Community-Based Epidemics (ESSENCE). However, both this data and the HHS data to which the reviewer referred is de-identified and aggregated at the hospital level. Therefore, these datasets include both Chicago residents and non-Chicago residents, the latter of which represents a considerable fraction of admissions to Chicago hospitals. CDPH datasets obtained from IDPH through I-NEDSS are filtered for Chicago residents only, regardless of the Illinois hospital to which they are admitted. If Chicago-resident-only data were available on a more frequent basis such that the reporting delay were only 1-2 days, as opposed to 5 days, outpatient sentinel surveillance would retain an operational recency advantage of about 5-6 days.

This description has been added to the Discussion section (L317-38).

Analysis suggestions

I agree with the authors that low sample size and noise in the sentinel case samples may be obscuring the lead time they might provide against hospital admission, though (<https://www.pnas.org/doi/10.1073/pnas.2111870119>) suggests there may be little lead time between case counts and hospital admissions. One thought in looking at the data is that the sentinel sample counts may be a good indicator in themselves (not just the positives). Have the authors explored that as a potential predictor? It seems similar to CLI, in that it captures a metric of symptomatic infections in the region, even if individuals don't ultimately test positive, and could provide a useful indicator to compare in Figure 4 and the supplemental analyses.

We have added sentinel samples as a time series to Figure 4 and supplemental Figures S6, S7, S8, S9, and S10. We find that $R(t)$ derived using sentinel samples as an indicator produced slightly worse agreement with hospital-based indicators than $R(t)$ derived from sentinel cases. These observations have been added to the Results section (lines 160-62).

It is interesting that the case counts provide little lead time for the hospital admissions, but I believe the authors could do a bit more to more fully explore the data. For example, the critical time periods for public health decision making generally fall to detecting a pandemic surge and detecting the peak of a

pandemic. Do the sentinel samples/cases provide lead time to these specific time periods over the course of the pandemic? One way to investigate this would be to identify the timing of the inflection points in the time-series (start and end of detected exponential growth, derivative changes, etc.), and see how the timing compares across the time series for these specific events rather than comparing the cross correlations across the full time-series.

Three major inflection points occurred during the study period: the peak of the Fall 2020 wave (Nov 2020), the valley preceding the Spring 2021 wave (Feb 2021), and the peak of the Spring 2021 wave (Mar 2021). The timing of these inflection points, obtained from the $R(t)$ traces of each indicator, are shown in Table S1. The dates of these inflection points in the sentinel cases $R(t)$ curve fall between 24 days behind to 11 days ahead of traditional indicators, which we now note in the Results section (lines 162-5). Additionally, we have added vertical lines representing dates where each $R(t)$ trace crosses 1.0 to Figure 4A.

Other studies have found strong geographic determinants for risk during the pandemic, and it appears that the authors have geographically stratified data. Are there any interesting features looking at the correlations across testing locations or comparing the sentinel geographic trends with the geographic trends in hospital admissions/other metrics?

Because individual testing sites weren't consistently active over the study period, this dataset would be of limited use for comparing temporal trends across geographies. This would certainly be an interesting and feasible analysis under a deliberate sentinel surveillance system with higher testing volume.

The authors mention and show the demographic data in Figure 3, but it appears that these data aren't used during the analyses. I think it would be useful to provide some results along these axes. For example, is test positivity highest in one race/ethnicity or age group compared to the others? The authors mention other studies that identified those trends, but it would be useful to see if similar results are obtained from the sentinel surveillance program.

Sentinel positivity rates by age and race\ethnicity, aggregated over the study period and week-by-week, are now shown in Figure S4 and discussed in the Results section (lines 129-130). Sentinel test positivity rates were highest in Hispanic/Latino patients, reflecting the finding in Illinois (Ref 13 and Ref 17) and elsewhere in the United States (Ref 15) that Hispanic/Latino residents are generally among the most undertested for SARS-CoV-2.

I think it would be useful to see all of the time-series of the indicators aligned with one another in some plot (maybe a supplemental figure). That would allow readers to get a better handle of the raw data underlying the R_t and correlation results.

We have added this plot as Figure S6.

Reviewer #2 (Remarks to the Author):

This is a very well presented and thoughtful study detailing the advantages of using sentinel surveillance of outpatients for epidemic inference, which (if connected correctly with fast action policy making) could

have given policy makers vital extra days to evaluate epidemic trajectory and assess current intervention suitability. I have a couple of what I'm calling major comments, but this is mainly driven by interest and I think would add to the paper overall but are not flaws of the current methods. These relate to adding a continuous assessment of sentinel data rather than discrete ($R_t > 1$ or < 1), which is also very useful for policy decisions from experience, and also exploring potential drivers of inaccurate R_t estimates.

We thank the reviewer for their positive comments.

Comment 1: In the operational recency evaluation (by the way this is really lovely phrasing for this - really clear) and in the similarity matrices used in Figure 4B and in the SI, you currently evaluate each data type by comparing if it correctly infers $R_t > 1$ or $R_t < 1$ in agreement with another data source. While it is important to know directionality of the epidemic, having a good estimate of continuous R_t is also important, as it also directly helps predict for example how many days until hospital beds become limited, which is helpful for planning and resource management. As a result adding a continuous similarity matrix (mean difference between median R_t estimates across all dates for example) would be helpful to know how accurate the sentinel data source is.

Spearman correlation matrices between each \$R(t)\$ trace are now available in supplemental Figures S7, S8, S9, S10, S12, and S13. This continuous evaluation closely qualitatively matches the results shown by the discrete agreement metric. This is noted in the Results section (lines 182-3 and lines 200-1).

Comment 2: Over half of the sentinel samples are from static sites, however, the contribution of each of the static sites to total sentinel samples changes over time, e.g. Harwood Heights contributes only in 2021. Is this because the static sites only became active at certain points in time? Or because of where the epidemic was spreading? Please add this comment as well as perhaps plotting in the SI the incidence of cases in each of the static sites over time in comparison to the Chicago incidence series as a whole.

Each of the ten sites shown in Figure 2 are static sites (fixed location, operational on a set schedule, as opposed to mobile sites, which were typically one-day pop-up events). No static testing sites except one (Gately Park, site c) were operational over the entire duration of the study period (now noted in Results L97-8). Under a deliberately designed sentinel surveillance system, testing sites would ideally remain consistently open and retain consistent hours. As we note in the Introduction (lines 65-70), this study opportunistically used data from community-based testing sites that had been established to improve access to testing in underrepresented groups, not to perform sentinel surveillance. We have added time series of sentinel sample and case counts by testing site as supplemental Figure S2.

Minor Comments:

Line 48-55: This is an important point that hospitalisations and deaths are more likely to be older individuals. I am not sure I agree with the phrase "better represent" here though and how it connects to being uninformative of transmission patterns in younger ages. The issue is that because severity increases with age, you have more statistical power to discern transmission in older ages than you would in younger ages just due to sample size. Practically speaking, you will not be able to detect changes in younger populations, but likely because there are not enough admissions to be statistically confident in a change rather than it being uninformative. Perhaps change to express that if transmission

trends are different in younger populations that it will be harder to detect from hospitalisations and deaths due to the age gradient in severe outcomes.

These remarks have been clarified (lines 48 – 53).

Figure 1: Really clear graphic. Just one comment - Perhaps add to legend that today is the Rt evaluation date (or change in figure) to get it to align with Figure 5.

The label for “today” in Figure 1 has been changed to “R(t) evaluation date”.

Table 1: Could you add “Deaths” as an indicator and associated delays. You talk about deaths in your introduction and reference this table so I was surprised to see deaths not in here.

A row for death counts as an indicator has been added to Table 1.

Line 88: After reading the introduction I was surprised to read that the majority of the testing sites were mobile, because after reading the introduction this surprised me as in lines 56-50 it says “testing criteria and sampling effort on the sentinel population are predefined and do not change with time”. Perhaps just worth mentioning in the introduction that sentinel sites despite being mobile may still be informative of transmission.

Although a majority of testing sites were mobile (167 out of 177), the vast majority of sentinel samples were collected at static sites (12,879 out of 13,952). The reviewer’s suggested note has been added to the Introduction (lines 63-4).

Figure 2A: It is quite hard to work out percentages from this plot when you have both Hispanic/Latino and Non-Hispanic Black populations, i.e. where there is a mix of the blue and red (e.g. Auburn Gresham i) in 2A). If the plot is trying to just convey that there are regions with high % Hispanic/Latino, high % Non-Hispanic Black, low % either and mixed regions, maybe a 4x4 bivariate color scale may be clearer and you can then plot the full 4 by 4 color scale.

We have split Figure 2A into two maps for visual clarity.

Line 103: Can also add reference to Figure 2B here which shows the relationship between sentinel cases and sentinel samples.

This reference has been added (line 109).

Line 112-119: Were any of these statistically different to the population as a whole? Given the denominators it seems likely that they are, but it would be good to add a statistical test here to convey that the sentinel samples are testing a significantly different proportion of the population.

Because this section involves many pairwise comparisons, we only added two-proportion Z-scores and associated two-tailed p values to the four comparisons mentioned in-text, all of which show a significant difference in proportions (lines 125-9).

Line 130: Please change CLI to COVID-like Illness (CLI) as this is the first time it has been used.

We expanded the acronym, thank you for spotting this.

Figure 4: Perhaps it would be helpful to show as another column here or linked to another SI figure, the count data underpinning the R_t in Figure 4A. This might help guide the reader a little bit more by the time they get to Figure 4C in terms of understanding the lead times etc. (It took me quite a few times to read through this section to be able to understand all the information that is contained in Figure C, which I think would have been much more obvious with the count data also shown.)

We have added this plot as Figure S6.

Figure 5: Could you add that shaded regions in B are 95% CI (presumably). Similarly what are the error bars in Figure 5C.

We added these descriptions to the figure legend (lines 244 and 246).

Lines 192-197: The text here is very informative for explaining the advantages and makes the message very clear. It is hard though to directly relate this to Figure 5. Perhaps it would be helpful to have COVID admits shown here.

We have added a line to Figure 5B that shows the evaluation date when COVID admits first returned $R(t) > 1$.

Discussion: Really well written and good paragraph discussing the lack of lead time between sentinel cases and hospital-based indicators.

Thank you!

Line 300: UK Zoe app also a good reference here showing that actually, UK population provided symptom data has been very informative of trends and has aligned well with more formal structures such as the UK ONS.

This is a great point, thank you! We have added this reference (line 358-61).

Reviewer #3 (Remarks to the Author):

The authors of the manuscript "Tracking changes in SARS-CoV-2 transmission with a novel outpatient sentinel surveillance system in Chicago, USA" describe the setup of a primary care outpatient targeted monitoring system to specific areas and compare the usefulness and added value with comprehensive systems as well as severe disease monitoring systems.

The study is well conducted and provides good insight as well as proposes strategies to focussed sentinel monitoring away from comprehensive testing and monitoring strategies.

Thank you for your positive comments.

A few comments to clarify some points are added below:

In the study it is assumed that the risk of hospitalisation is equally distributed across all age groups to be

able to compare between primary and secondary care. However, the risk to be hospitalised increases significantly with increasing age and with the higher proportion of younger age groups in the sentinel the risk to be hospitalised might not be comparable to data from hospital settings. This could add bias to the results when comparing different age groups or risk groups across the different data sets. Have you tried to just use the 60+ age group for the comparison? Or use an age stratification? This might help to account for this risk difference in the sampled populations.

We have added an analysis that splits indicators, including sentinel cases, into ages >60 and <59, and then recalculates $R(t)$ using these age-stratified groups. This analysis is shown in Figure S10. We find that $R(t)$ derived from sentinel cases still agrees very well with other indicators under age 60, which suggests that SS competes with hospital-based indicators even in age groups with lower risk of severe outcomes. Agreement between sentinel cases ages 60+ and other indicators ages 60+ is low, which is expected given the low volume of sentinel samples in this age group (only 9.9% of sentinel samples were greater than 60 years old). These results are noted in the Results section (lines 168-71).

We should note that we do not assume that the risk of hospitalization is equally-distributed across age groups, nor is this assumption necessary to compare sentinel cases as an indicator to hospitalizations. We make this comparison because COVID-confirmed admissions are commonly used as an indicator for SARS-CoV-2 transmission in the general population, in spite of the highly disparate age distribution of admissions versus the population at-large.

It is unclear if there might be a bias by the inclusion of those with known symptom onset in the sentinel group. Have you tried a sensitivity analysis that compared the included and excluded groups in terms of demographics but also on test performance/positivity including over time?

We have now assessed demographic breakdowns and positivity rates at each level of selection of sentinel samples: all tests collected at sentinel sites, all tests collected at sentinel sites with Chicago residence, all tests collected at sentinel sites with Chicago residence and valid date of symptom onset, and sentinel samples (see Figure S1). We find that each of these groups were highly demographically similar (see Figure S5). The only large demographic difference between the two is that Non-Hispanic Black Chicago patients were more frequent among sentinel samples (25.8%) than among all tests collected at sentinel sites with Chicago residence and valid symptom onset (21.7%) and among all tests collected at sentinel sites (16.6%). We now note this in the Results (lines 129-30). This reflects the potential for demographic bias in the process of selection of sentinel samples, which we now note in the Discussion (line 258-260). Across the study period, sentinel samples showed a higher test positivity rate than all tests collected at sentinel sites (see Figure S3). This is to be expected, since the selection of sentinel samples screens for symptom onset, and symptomatic patients are more likely to harbor a SARS-CoV-2 infection.

Are data from people not included into sentinel due to lack of symptom information included in the overall dataset? Are the sentinel data included in the total case data? Please add for clarity.

Sentinel samples form a subset of all tests collected at sentinel sites, which in turn form a subset of all diagnostic tests. Sentinel cases form a subset of all cases. We note in the Results section that 0.4% of all diagnostic tests were also sentinel samples (lines 133-4). In the caption of Figure 3, we added a note clarifying that sentinel samples were a subset of all diagnostic tests (line 140).

It is shown that sentinel test positivity performed worse than just pure case count, which is quite surprising. While simple case count is dependent from number of tested people (and proportion testing positive), this might be influenced by different testing strategies over time. Could you give some more details about this (e.g., in the supplement). This has not been discussed yet and seems remarkable as TPR is a commonly used indicator (for week positivity in flu surveillance systems). Have you tried using weekly data with positivity? Why do you think TPR does not describe the epidemiological situation well although having denominator data available? Could you reflect a bit more on this please? Fig S3 shows a much better performance of TPR in the later period than in the earlier. This is an interesting finding and could be addressed in the discussion.

Multiple factors might explain why sentinel TPR does not perform as well as raw sentinel case counts. From what we have observed previously from regular SARS-CoV-2 outpatient diagnostic testing data, TPR can go up independent of incidence because of decreases in test availability or in test demand, for example during the snowstorms in Chicago in February 2021 when access to testing was physically more challenging. This is because in periods where testing is more restricted, only those that are most likely to test positive (e.g. known prior contact or actively symptomatic) are likely to access a test. Conversely, TPR may also fall independently of incidence if a large volume of diagnostic tests is suddenly made available to a population with limited access to testing. These points are now addressed in the discussion (lines 264-73). In influenza surveillance systems, diagnostic testing is far less demographically, geographically, and temporally heterogeneous, leading to better consistency for TPR as an indicator. Such observations have also been made by others (see Refs 1, 13, 14, 15, 16, 17). It is possible that the low agreement from TPR in the early study period is due to the more frequent changes in test site hours and locations.

It has not been described if this sentinel system would become permanent or just served temporarily. No outlook has been added in terms of the usefulness of (temporary) sentinel systems to monitor the situation of SARS-CoV-2 and also include other respiratory viruses. Could you please elaborate a bit more on this how such a system could be used and embedded to complement other systems and when it would be most beneficial.

The sentinel surveillance described in this manuscript was temporary. A deliberate sentinel surveillance system would be continuously useful for its advantage in operational recency and could operate on a temporary or permanent basis (regardless, the same requirements of consistency would be needed). Such a surveillance system would be most useful in situations with:

- Low outpatient testing availability (as symptomatic individuals would be more likely to seek a test).
- Low rates of severe outcomes brought on by high vaccination rates (newer variants still cause symptomatic breakthrough infections).

This is now described in the Discussion (lines 363-66).

I'm no expert in nowcasting and cannot assess the suitability of this approach although the results are promising. How well and timely could such nowcasting be implemented in ad hoc systems or newly established sentinel system and how much has nowcasting been used for public health decision making during the time of the sentinel system implementation in Chicago.

Nowcasting has been applied extensively during the COVID-19 pandemic (see Refs 32 – 36) and during past disease outbreaks (see Ref 37) to adjust for right-censoring and undercounting of case counts. It is presently deployed by the UK Health Security Agency to adjust for right-censoring in the current monkeypox virus outbreak (see <https://www.gov.uk/government/publications/monkeypox-outbreak-technical-briefings/investigation-into-monkeypox-outbreak-in-england-technical-briefing-4>). At a high level, nowcasting could be parameterized and deployed dynamically based on the latest available retrospective data from the sentinel surveillance system, as shown in Figure 5C. So long as a public health department kept past datasets, this approach is feasible. There are some challenges associated with nowcasting: nowcasting does not involve any kind of mechanistic understanding of testing or reporting practices and could produce erroneous estimates if there are unexpected shifts in testing or reporting behavior. Moreover, testing and reporting practices may vary by public health agency, and a nowcasting approach used in one locale may not suffice for correcting for reporting practices in a different locale. Nowcasting was not employed during the study period in Chicago, but is now used to adjust for recent undercounting in hospital admissions.

It's said that a PCR was performed for sentinel sites, but it is unclear if the tests have been at sentinel sites or had to be transported to lab facilities and how the reporting to patients and health authorities had been conducted. Could you please add some more details?

Specimens were collected at testing sites and transported to an off-site laboratory for processing via PCR. Testing vendors notified patients of results electronically as soon as results were available. During the study period, the median turn-around time from specimen collection to result notification was 2 days, with 95% of tests turned around between 1 and 4 days. Data for all available test results were made electronically available to CDPH and IDPH on a daily basis. This information has been added to the Methods (lines 405-9).

Sentinel systems for influenza are considered the gold standard as they are set up to be representative of the population to monitor the activity, spread and the circulating viruses in a timely manner. Sentinel flu viruses are sent to National Influenza Centres where viruses are subtyped, genetically and antigenically characterisation. This provides data for vaccine composition and delivers subtype information that is not available for most other specimens tested in non-sentinel diagnostic laboratories (at least in Europe). Sending specimens to central reference laboratories delays results which might not be relevant for influenza, but for SARS-CoV-2 particularly when a new variant emerges that needs to be closely monitored. The study does not mention any specific analysis of SARS-CoV-2 variant viruses among the positive tested specimens. It should be discussed how to integrate virus characterisation in such a sentinel system e.g., per specific drop-out PCR, sequencing, or more sophisticated analyses e.g., requiring virus culture.

Integrating sequencing and variant identification into a sentinel surveillance with consistent symptom data collection would have the added benefit of rapidly providing information on symptom presentation of new variants. This point has been added to the Discussion (lines 361-3).

REVIEWERS' COMMENTS

Reviewer #1 (Remarks to the Author):

I have reviewed the new version of the manuscript and the author's responses to reviewer feedback. I appreciate the author's significant efforts in revising the manuscript and believe it to be much improved! I congratulate the authors on writing a clear and impactful paper and have no further comments.

Reviewer #2 (Remarks to the Author):

Thank you for your thorough review. All my points have been addressed sufficiently and clearly.

Reviewer #3 (Remarks to the Author):

My comments and suggestions have been satisfactorily addressed and included, nothing to add otherwise. Many thanks